# Single-Dose P2 X4R Single-Chain Fragment Variable Antibody Permanently Reverses Chronic Pain in Male Mice

**DOI:** 10.3390/ijms222413612

**Published:** 2021-12-19

**Authors:** Karin N. Westlund, Marena A. Montera, Aleyah E. Goins, Sascha R. A. Alles, Nikita Suri, Sabrina L. McIlwrath, Robyn Bartel, Ravi V. Durvasula, Adinarayana Kunamneni

**Affiliations:** 1Department of Anesthesiology & Critical Care Medicine, University of New Mexico Health Sciences Center, Albuquerque, NM 87131, USA; monteram@salud.unm.edu (M.A.M.); alegoins@salud.unm.edu (A.E.G.); salles@salud.unm.edu (S.R.A.A.); rbartel1@salud.unm.edu (R.B.); 2Biomedical Laboratory Research & Development (121F), New Mexico VA Health Care System, Albuquerque, NM 87108, USA; sabrina.mcilwrath@va.gov; 3Ross University Medical Center, Miramar, FL 60153, USA; NikitaSuri@mail.rossmed.edu; 4Department of Medicine, Mayo Clinic, Jacksonville, FL 32224, USA; durvasula.ravi@mayo.edu (R.V.D.); Kunamneni.AdiNarayana@mayo.edu (A.K.)

**Keywords:** small antibody, chronic pain, nerve injury, neuropathy, hypersensitivity, anxiety, depression, pain therapy

## Abstract

Non-opioid single-chain variable fragment (scFv) small antibodies were generated as pain-reducing block of P2X4R receptor (P2X4R). A panel of scFvs targeting an extracellular peptide sequence of P2X4R was generated followed by cell-free ribosome display for recombinant antibody selection. After three rounds of bio-panning, a panel of recombinant antibodies was isolated and characterized by ELISA, cross-reactivity analysis, and immunoblotting/immunostaining. Generated scFv antibodies feature binding activity similar to monoclonal antibodies but with stronger affinity and increased tissue penetrability due to their ~30% smaller size. Two anti-P2X4R scFv clones (95, 12) with high specificity and affinity binding were selected for in vivo testing in male and female mice with trigeminal nerve chronic neuropathic pain (FRICT-ION model) persisting for several months in untreated BALBc mice. A single dose of P2X4R scFv (4 mg/kg, i.p.) successfully, completely, and permanently reversed chronic neuropathic pain-like measures in male mice only, providing retention of baseline behaviors indefinitely. Untreated mice retained hypersensitivity, and developed anxiety- and depression-like behaviors within 5 weeks. In vitro P2X4R scFv 95 treatment significantly increased the rheobase of larger-diameter (>25 µm) trigeminal ganglia (TG) neurons from FRICT-ION mice compared to controls. The data support use of engineered scFv antibodies as non-opioid biotherapeutic interventions for chronic pain.

## 1. Introduction

### Targeting P2X4R with an scFv for Chronic Pain Reversal

P2X4R is an ATP-gated purinergic receptor ion channel found sparingly in the cell membrane under normal conditions but not typically involved in nociception itself [1]. After nerve/tissue injury, P2X4R stored in late endosomal, lysosomal and/or lysosome-related organelles near the membrane is released by lysosomal ATP and fuses with microglial cell membranes for rapid response [2,3,4,5,6,7,8]. Upregulation of P2X4R is measured within 1 h after nerve injury, trafficked to the spinal microglial membrane, and contributes to mechanical allodynia [1,3,9,10,11,12,13,14,15]. ATP-gated Ca^2+^ influx through P2X4R leads to phosphorylation and activation of p38-MAPK [16]. As a result, spinal microglia produce and release brain-derived neurotrophic factor (BDNF), initiating persisting activity in second-order neuronal cells and the disruption of neuronal Cl^-^ homeostasis causal in persistence of pain and opioid tolerance [3,17,18,19,20,21]. Both phosphorylated p38 and P2X4R are initially exclusively co-localized with the microglial marker, OX-42, in the first three hours after injury, and not with neuronal or astrocyte markers [15,22]. With continued activation, P2X4R are widely expressed in peripheral nerves and in neurons and astroglia (GFAP) throughout the central nervous system (CNS). The resulting central sensitization chronifying neuropathic pain requires activation of astrocytes [18,23]. Increased cell membrane expression of P2X4R after nerve injury is only detected in male rats [24]. In females, P2X4R is not involved in the microglial activation pathway mediating pain hypersensitivity [25,26,27,28].

Peripherally, P2X4R is found in activated T cells, B cells, tissue-resident immune cells such as macrophages, and in various epithelial tissues and endothelial cells. Significant immune cell infiltration is found in the dorsal root ganglia (DRG) compared to spinal cord. In male animals with spinal nerve ligation, B cells are more prominent in males 8 days after insult, and T cells are more prominent in females [26].

The present study details a novel approach for a permanent reduction in chronic nerve injury induced hypersensitivity in male mice targeting P2X4R (Figure 1). Single-chain variable fragment (scFvs) antibodies are a truncated type of antibody. Consisting of only a portion of the binding arm, they are only approximately 30% the size of IgG molecules but with retained binding affinity comparable to monoclonal antibodies (Figure 2). Our scFvs are generated via ribosomal display, a powerful cell-free technology widely used to select scFv antibodies against the target of choice particularly for cancer therapy. Our scFvs demonstrate stronger tissue and brain permeability, compared to antibodies developed via the more difficult phage display technology typically. Generation of selected scFv in sufficient quantities for animal testing is achieved with *E. coli* Rosetta-gami cytoplasm expression and purification. This method has been used by us to quickly develop repertoires of high-affinity antibodies.

Proteins targeted by our group are recombinant products of RNA that are upregulated in chronic pain models, but not involved in acute nociception. The P2X4R scFvs have high affinity, superior stability and solubility as with scFvs we previously developed against Zika virus, filovirus glycoproteins, and cholecystokinin B receptor [29,30,31].

Further innovation demonstrated in this study includes a sex-specific permanent reversal of chronic neuropathic pain with our newly engineered P2X4R scFv antibody administered only once 3 weeks post-induction of a chronic trigeminal nerve injury model. The preclinical Foramen Rotundum Inflammatory Constriction Trigeminal InfraOrbital Nerve (FRICT-ION) model was used that causes ongoing mechanical hypersensitivity for at least 15 weeks as well as secondary anxiety- and depression-like behaviors after week 3–6 [31,32]. The data described here find a single dose of our P2X4R scFv permanently restores pain-related behaviors to baseline in male mice with chronic FIRCT-ION pain, while in treated female and untreated chronic pain model mice the symptoms persist through the 10week time course. Development of anxiety- and depression-like behaviors was prevented in P2X4R scFv-treated FRICT-ION mice. Characterization of direct effects of P2X4R scFv on neuronal activation responses was examined in primary trigeminal ganglia (TG neuron) cell cultures from the male mice 3 weeks after FRICT-ION-induced neuropathic pain. The P2X4R scFv decreased neuronal excitability. It is clear that complex, multifactorial maladaptive mechanisms are responsible for maintaining neuropathic pain in the long term, and thus novel approaches are warranted to quell the changed neural and immunological environment of chronic pain.

## 2. Results

### 2.1. Generation of P2X4R scFv and Specificity of Binding

Extracted RNA from full-length antibody was isolated from five mouse spleens after three immunizations with a unique custom 13 a.a. extracellular sequence of P2X4R (Figure 1). The peptide has 11/13 a.a. residues identical to human and 12/13 identical to mouse. cDNA libraries that encoded the immunoglobulin heavy and light chain variable regions (V_H_ and V_L_) were constructed for ribosome display. cDNA libraries that encoded the immunoglobulin heavy and light chain variable regions (V_H_ and V_L_) were constructed for ribosome display (Figure 2A). Three (3) rounds of panning of the ribosome-displayed scFv library against the P2X4R peptide were performed. PCR products cloned into the pGEM-T Easy vector, transformed in *Escherichia coli* cells and approximately 100 clones of V_H_—V_L_ transformants were later randomly selected for sequencing. Following sequencing, the anti-P2X4R recombinant scFvs were further cloned into the expression vector, pET32a. The scFvs were expressed and purified from *E. coli* cytoplasm as described previously for the generation of antibodies against the Zika virus and filovirus glycoproteins [29,30], and CCK-B receptor [31]. Western blot of the cytoplasmic extracts showed specific detection of the soluble scFvs by the mouse anti-His tag antibody (Figure 2B). The binding activity of the soluble scFvs was measured by indirect ELISA (Figure 2C). A total of 10 clones showed binding activity with P2X4R peptide, with clones 12, 95 and 103 showed the highest binding activity while no cross-reactivity with CCK-BR receptor protein or the negative control anti-Zika scFv 7-2 [29,30] (Figure 2D). The aligned amino acid sequences of these three scFvs using Clustal Omega are shown in Figure 3. The panning process was efficient in selecting clones of high affinity (Figure 2C) as was evident from the affinity differences of the scFvs: scFv95, scFv12, and scFv103 had the highest and second highest affinities, respectively, while others had lower affinity [30]. The scFvs with the highest affinities (95 and 12) were selected for in vivo behavioral testing.

### 2.2. In Vivo Behavioral Characterization of Two P2X4R scFvs in Chronic Pain Models

#### 2.2.1. P2X4R scFvs Permanently Reverse Mechanical Hypersensitivity in the FRICT-ION Model

The FRICT-ION model persists at least 105 days (15 weeks), with mechanical hypersensitivity remaining significantly different from naïve at all weekly time points (*n* = 4; F(15,144) = 284.3, *p* < 0.0001, two-way ANOVA) of chronic clinical neuropathic pain. Thus, the model is relevant for testing pain therapeutic candidates.

A single dose of P2X4R scFv (4 mg/kg, i.p.) permanently reversed mechanical hypersensitivity in the FRICT-ION model male mice to baseline. After three weeks of persisting mechanical hypersensitivity, the P2X4R scFv95 and scFv12 were both efficacious (Figure 4A), but this was not replicated in female mice known to have a different P2X4R activation mechanism (Figure 4B) [25,26,28]. Two-way ANOVA (Dunnett’s multiple comparisons test) was performed on these data (*n* = 6, F (24,195) = 435.7). In post hoc analyses, Bonferroni adjustment to all *p* values for week-by-week comparisons of FRICT-ION versus vehicle control yields all nine *p* values < 0.001. Only data for male mice were collected after this result.

Block with a peptide inhibitor recognizing the target P2X4R sequence provides only a short-term increase in the mechanical threshold, significant only at the 4 h time point (Figure 4C), but efficacy did not persist as it did with the P2X4R scFvs.

#### 2.2.2. P2X4R scFv95 Prevents Anxiety- and Depression-like Behaviors

Efficacy for a reduction in anxiety-like behavior. Based on its efficacy to reduce reflexive mechanical hypersensitivity in animals with persistent pain induced by the FRICT-ION model, the P2X4R scFv95 was selected as a lead candidate for additional testing of emotion-like behavioral assays. Vehicle-treated animals with FRICT-ION model-induced chronic pain displayed anxiety-like behavior. With the light/dark test, the number of transitions into and occupancy of the light chamber were significantly reduced (Figure 5A,B) compared to naïve controls. The time delay until vehicle-treated FRICT-ION mice entered the light chamber was significantly increased (Figure 5C) and exploratory rearing events decreased (Figure 5D) compared to naïve controls. Treatment of male FRICT-ION mice with P2X4R scFv95 significantly reversed these measures so that they were not different from naïve control animals (*n* = 6, one-way ANOVA, Dunnett’s multiple comparisons test; ** *p* < 0.01 and *** *p* < 0.001) (Figure 5). P2X4R scFv 95 was efficacious in reversing all the measures which were not significantly different from naive controls.Efficacy for a reduction in depression-like behavior. Depression-like behavior was determined in week 5 post-FRICT-ION model induction using the sucrose splash test. The number of grooming events and total time spent grooming were significantly reduced in vehicle-treated FRICT-ION mice while the time delay until the first grooming event was increased when compared to naïve control animals (Figure 6A–C). Treatment of male FRICT-ION mice with P2X4R scFv95 significantly increased the number of grooming events so that they were not different from naïve control animals (Figure 6A). However, neither their total time spent grooming (Figure 6B) nor their delay until the first grooming event (Figure 6C) were significantly different from vehicle-treated FRICT-ION mice.

### 2.3. Evidence for Brain Penetrance by P2X4R scFv

Evidence for peripheral sensory neuron ganglia and brain penetrance of the P2X4R scFv 95 was sought with Western blot of tissues dissected week 7 after His-tagged P2X4R scFv95 administration. The His-tag was detected in amygdala, medullary brainstem, and TG of mice with FRICT-ION (Figure 7A). No His-tag was detected in the medial pre-frontal cortex (mPFC). The bar plot provides comparison of the content (Figure 7B).

### 2.4. P2X4R scFv95 Reduces Excitability of TG Neurons from FRICT-ION Mice In Vitro

Whole-cell patch-clamp electrophysiological recordings were performed on primary cultures of TG neurons obtained from FRICT-ION mice (3–4 weeks post-injury) in the presence and absence of P2X4R scFv95 (4.5 μg/mL,1–2 h pre-incubation). Figure 8A shows that there were differences in the distribution of single and multiple firing neurons in untreated vs. treated TG neurons with a higher prevalence of multiple firing neurons in untreated controls. However, a Fisher’s exact test did not reveal any statistically significant differences in these distributions (*p* = 0.3245). In Figure 8B, we compared the proportion of rebound firing neurons observed between untreated and treated neurons. Rebound firing neurons are correlated with cold-sensing properties of nociceptors33. We noted that there were 4/19 in untreated conditions (~21%) and 1/18 in P2X4R-scFv-treated conditions (~6%). A Fisher’s exact test comparing numbers of rebound and non-rebound firing neurons between untreated and treated conditions did not show significance (*p* = 0.3398) and this is most likely due to the relatively low prevalence of rebound firing neurons observed in our cultures. Finally, we analyzed rheobase (current injection required to elicit firing), resting membrane potential (RMP) and maximal action potential (AP) firing evoked by stepwise current injection. We separated neurons into either large diameter (>25 microns, Figure 8C) or small diameter (<25 microns, Figure 8D). P2X4R-scFv increased rheobase significantly (*p* < 0.05, unpaired *t* test with Welch’s correction) in large-diameter cells, but did not have a significant effect on RMP (*p* = 0.9580, Welch’s *t*-test). There was a noticeable reduction in maximal AP firing of large-diameter neurons, but this was not statistically significant (*p* = 0.0691, Welch’s *t*-test). There were no significant differences between untreated and P2X4R scFv-treated small-diameter neuron rheobase, RMP or maximal number of Aps. These results show that P2X4R scFv reduces excitability of large-diameter TG neurons by increasing their rheobase, which may underlie the anti-allodynic action of the scFv.

### 2.5. Pre-Block of P2X4R Immunostaining by scFv95 Antibody

Immunofluorescent staining for P2X4R was minimally evident on TG neurons in primary cultures isolated from naïve BALBc male mice (Figure 9A top,B). Quantified staining intensity was significantly greater in TG primary cultures isolated from FRICT-ION mice compared to those from naïve mice as illustrated in the images and the bar graph (Figure 9A bottom,C). In TG primary cultures pre-blocked with increasing concentrations of P2X4R scFv95 antibody (0.5, 5, 10 µg/mL, 24 h), immunocytochemical staining intensity of P2X4R was incrementally less as concentration of P2X4R scFv95 increased in TG from both naive (Figure 9D) and FRICT-ION mice (Figure 9E).

## 3. Discussion

These data open new avenues of research for development of scFv therapeutic interventions for chronic pain. Data presented indicate efficacy for a reduction in mechanical hypersensitivity, and the specificity, efficacy, and brain penetrance of the P2X4R scFv95 with the potential to progress as therapy for chronic pain in males. The clear sex difference, however, emphasizes the importance of finding other potential therapies for females. The innovation of these data includes successful permanent reversal of pain-related behaviors the mouse model of chronic trigeminal pain using P2X4R-specific scFv antibody therapy with a single treatment in male mice though having no effect on female mice. The scFv antibody block of P2X4R promoted long-term restoration of baseline mechanical threshold within a week and prevented anxiety- and depression-like behavior that develops in the untreated mice. A peptide blocker targeting the same sequences in P2X4R as well as current analgesics only temporarily reduce pain. Since P2X4R is present in low levels under normal conditions and increases after nerve injury [3], targeting P2X4R in chronic rather than acute models has been particularly successful in this study. The likelihood of interference with altered nerve function in the persisting FRICT-ION model is evidenced by significant reduction within one week, full return of mechanical hypersensitivity behavioral responses to naïve baseline within two weeks, and a reduction in neuronal excitability at the single cell level with the patch-clamp recordings in TG neuronal cells isolated from mice with FRICT-ION hypersensitivity.

Consideration of targets found to be upregulated with gene profiling at the transition to chronic pain has thus far been limited. Our use of chronic pain models in our studies provides translational relevance. Understanding the role that blocking P2X4R in the long term would have on nociceptive circuitry relevant to the pain-related behaviors was tested. The use of electrophysiological characterization aided in the understanding of the direct effects of P2X4R specific scFvs on afferent neurons. Primary sensory neuron excitability in vitro shown here provides a potential mechanism relevant to the block of chronic pain in the mouse model. The role of large-diameter neurons such as those implicated in cold sensing and Aβ nerve fibers are important in chronic pain behaviors including spontaneous pain and tactile allodynia [33,34,35]. The findings here imply the P2X4R scFv95 has potential to block phenotypic switching from acute to chronic pain. Further study is needed to test this assertion.

### 3.1. Comparison of Monoclonal versus scFv Antibodies

Antibody therapy development for management of pain has exploited the high selectivity of conditioned monoclonal antibodies (mAbs) to target a specific cell or protein type. Among antibody-based therapies and immune modulators—seven antibodies are currently in phase III trials that are targeted to reduce immune factors, including CGRP for migraine; semaphorin 4D (Huntington’s disease); nerve growth factor; tau, α-synuclein, and amyloid β (Alzheimer’s disease, ALS, Parkinson’s disease), demonstrating current thinking and the diversity of available targets for immunotherapy in neurological disorders [36,37]. Despite the success of mAbs, the development process is laborious, expensive, and succumbs to the difficulty of generating antibodies against self-antigens in humans. Full-length rodent antibodies can cause systemic inflammation in humans and are rapidly removed from circulation limiting their clinical application. Humanized scFv antibodies circumvent these issues.

In recent years, in vitro display techniques, including phage and ribosome display, have become platform technology for the design, selection and production of reagents for targeted therapies, including for P2X3R [38]. Phage display has limitations including the inability to select antibodies under conditions different from the cell environment, problems with the selection of proteins that are toxic, cells circumventing selection pressure, and low transformation efficiency. All these deficits may be circumvented using in vitro ribosome display. Thus, the more favored cell-free ribosome display platform technology was utilized for the design, selection and production of reagents for a variety of targeted therapies [29,30,39,40,41,42].

Smaller engineered antibodies, including Fabs, scFvs, and diabodies, feature binding activity similar or better than monoclonal antibodies but with stronger tissue and brain penetrability. Penetrance is evidence by His-tag remaining in the brain samples after seven weeks. In the present study, penetrance is occurring without disruption of the blood–brain barrier other than what is occurring naturally as a result of the neuropathic pain model insult. Due to their solubility and small size, which allows them to cross the blood-nerve and blood–brain barriers, scFvs are being investigated as therapeutics for arthritis, Alzheimer’s, Parkinson’s, Creutzfeldt-Jacob, and Huntington’s disease [43,44,45]. Computational modeling allows predictive illustration of the binding to target proteins [46]. The scFvs can be easily modified with an in vivo half-life for short-term diagnostic or long-term biotherapeutic applications for both the nervous and immune systems. These antibody-based therapies have been applied as cancer treatment for a decade. The advantages, along with lower production cost, the ease of large-scale production due to the use of ribosomal display, provide evidence that scFvs can be utilized as a viable treatment method for chronic neuropathic pain.

The increased brain permeability evidenced in the TG, medulla, and amygdala by the Western blot indicated specific targeting, superior stability, and solubility providing the effective block of targets within the pain circuitry components. These qualities make scFv antibodies well suited for selective targeting of P2X4R and their further development for chronic pain treatment. These technologies can overcome previous challenges inherent in providing therapeutic applications for P2X4 [9,12,47,48]. The scFvs can be designed to target other use dependent receptors identified in the nervous system that upregulate or are imposed into the membrane after physiological challenge. Furthermore, the scFv antibodies have promising biotherapeutic applications for both the nervous and immune systems, which are recognized as interactive in chronic pain [43,45,49].

### 3.2. Potential of scFvs as Non-Opioid Chronic Pain Therapy

Transition to chronic pain is a serious consequence of nerve injury. Current understanding is that pain chronification involves physiologic, immunologic, molecular, epigenetic, and brain circuitry changes. Pain persisting after tissue healing, or “neuropathic pain”, remains a significant clinical challenge with a treatment response rate of only 11% [50,51]. Reliance on opioids post-surgically does not stem the transition to chronic high impact neuropathic pain occurring in 10% of patients [52]. Opioids as a therapeutic agent, in particular, are significantly limited by their addictive and detrimental side effects, including sedation, respiratory depression, constipation, and tolerance. In order to effectively treat chronic pain, without the risk of damaging side effects, non-opioid alternatives must be identified to alleviate the risk of dependence or addiction. Effective and specific therapeutics with higher efficacy and fewer side effects for chronic pain such as effective scFv antibody therapies are urgently needed [53,54].

P2X4R was identified as the primary channel activated through which pro-inflammatory cytokines and other active neuronal substances are subsequently activated for release following injury [55,56]. Activation of P2X4R has been shown to initiate calcium influx and p38 MAPK phosphorylation peripherally and centrally resulting in neuropathic pain [1,13,14,15,57]. In fact, the ATP induces spinal LTP by activation of P2X4 receptors 60 min after LTP induction [15]. At that time the level of phospho-p38 mitogen-activated protein kinase (p-p38 MAPK) is significantly elevated and at 180 min after LTP the number of P2X4R increases significantly [15]. Increased primary afferent firing in turn activates spinal dorsal microglia and tissue–resident macrophages to release TNF-α, IL-1β, IL-6, ATP, NGF, NO, ROS, prostaglandins, and other active neuronal substances and pro-inflammatory cytokines that participate in maintenance of neuropathic pain [1]. Partial nerve injury induces activation of P2X4R in the DRG causes release of substance P from damaged neurons and inflammatory substances from satellite glial cells [58]. In a rat model of neuropathic pain of tactile allodynia in which animals become hypersensitive to mechanical stimulation after peripheral nerve injury, Tsuda et al. (2003) found that intrathecal knock down of P2X4R via antisense oligonucleotides decreased hypersensitivity [12]. Furthermore, the selective P2X4R antagonist NP-1815-PX produces brief anti-allodynic effects in chronic pain models, which support the hypothesis that microglial P2X4R could be a potential target for treating chronic pain [48]. Unfortunately, female mice are resistant to antagonism of P2X4R and to our P2X4R scFv format. Another P2X4R scFv antibody developed that was converted back to an IgG is effective in female mice but the full-length IgG would have the inherent issue of reduced penetration of nervous tissue [59].

### 3.3. Evidence of Brain Pnentrance

As suggested by our Western blot data for the P2X4R scFv antibody His-tag remaining in the TG and brain at 10 weeks after the single-dose treatment (Figure 8), brain penetrance of the P2X4R scFv likely assisted in the reduction in pain-related behavior. It is unclear if the effect on mechanical hypersensitivity is due to block of the P2X4R’s action on primary afferent endings peripherally or centrally in the medulla since his-tag was present at both sites. The His-tag marker in the medulla may be accumulated from release by primary afferent endings there or may be evidence of brain barrier penetrance at that site. Whether block of afferent firing of the primary afferent nerves and/or the potential block by the P2X4R scFv in the medulla is providing the ability to return to baseline behavioral responses, is unclear at this time requiring additional study. The abundance of his-tag that was conjugated to the P2X4R scFv protein in the amygdala more clearly suggests brain penetration by the P2X4R scFv prevented the development of anxiety- and depression-like behaviors although diminished afferent firing was likely another major factor.

### 3.4. Potential of P2X4R scFv as a Pain Therapeutic

The finding that complete reversal of mechanical hypersensitivity is possible after the single dose of scFv is highly unusual since most analgesics revert back to previous pain levels within hours or days at most. Development of opioid-induced hypersensitivity with continued use is now well known [60,61]. These data are difficult to explain other than to suggest that natural healing processes are allowed to proceed as indicated by clues that lie among the RNAseq data in previous nerve injury studies. In one study indicating recovery of chronic nerve constriction injury in males it was suggested that an increased expression of P0 and Neu200 may contribute to the recovery in males (81 days) compared to females (121 days) [62]. Additional study is clearly needed to unravel the physiological/molecular mechanisms allowing complete recovery of the chronic pain state to baseline in the animal model. Clearly, the P2X4R scFv is providing promising potential for further development as a therapeutic for chronic pain.

## 4. Materials and Methods

### 4.1. Generation and Selection of P2X4R scFv Antibodies Using Ribosome Display

Total RNA was isolated from spleens of five mice immunized with a custom extracellular peptide sequence (a.a. 301–313, C-RDLAGKEQRTLTK, MW 1516 g/mol) of rat P2X4R with an N-terminal biotin tag (GeneScript, Piscataway, NJ, USA). The peptide has 11/13 a.a. residues identical to the human peptide and 12/13 identical to the mouse peptide sequence. cDNA libraries that encoded the immunoglobulin heavy and light chain variable regions (V_H_ and V_L_) were constructed for ribosome display as previously described for the generation of antibodies against the CCK-B peptide [31]. Three rounds of panning of the ribosome-displayed scFv library against the P2X4R peptide were performed. PCR products cloned into pGEM-T vector were used to transform *E. coli* competent cells and approximately 100 clones of V_H_—V_L_ transformants were later randomly selected for sequencing. Following sequencing, the recombinant anti-P2X4R scFvs were further cloned into the expression vector, pET32a. The scFvs were expressed and purified from *E. coli* cytoplasm as described previously for the generation of antibodies against the Zika virus and filovirus glycoproteins [29,30]. Binding, specificity, and affinity of the scFvs to target P2X4 peptide were determined by indirect ELISA as previously described for the generation of antibodies against the CCK-B peptide [31].

### 4.2. FRICT-ION Chronic Trigeminal Neuropathic Pain Model

Animal studies are reviewed and approved by the University of New Mexico Health Sciences Center Institutional Animal Care and Use Committee as IACUC# 220-201013-HSC (3 September 2021). The trigeminal nerve injury rodent models established in the literature for the study of trigeminal orofacial pain have been refined over the years and used by numerous labs in the field of pain research for study of the neuronal pathways and mechanisms causal in pain [63,64,65,66,67,68]. Our latest refinement, development of the Foramen Rotundum Inflammatory Constriction of the Trigeminal InfraOrbital Nerve (FRICT-ION) model, is a minimally invasive variant and rapid method useful for both rats and mice [32]. BALB/c white mice were used since they remain cooperative through the 10 weeks of behavioral testing. although C57bl6 can also be used in future studies with specific KO animals. One lip of anesthetized mice is secured with cotton suture to expose the buccal-cheek crease, where a tiny scalpel cut exposes trigeminal nerve roots innervating the teeth. A 3 mm section of chromic gut suture is slid along the infraorbital nerve into the foramen rotundum as it enters the skull. This rapid 5–10 min method produces hypersensitivity over the subsequent week that persists over the seven week experiment. Estimates are that in week 6, mice have experienced pain equivalent to 8 human years and can be considered chronic [69], making it ideal for testing potential non-opioid therapeutics at chronic time points. Control mice (sham-operated) undergo the same surgical procedure without nerve manipulation (Figure 4B). Naïve mice remain untouched.

### 4.3. Bioefficacy Testing of High-Affinity Anti-P2X4R scFv Antibodies

Dose response (0.04, 0.4 mg/kg) was assessed in both male and female mice in blinded studies. The scFv antibody optimal dose or vehicle intraperitoneal (i.p.) injection was given to mice (control, sham, nerve injured). P2X4R scFv antibodies were administered in week 3 post-FRICT-ION model induction, when anxiety-and depression-like behaviors were developing in mice with nerve injury.

A single injection of vehicle (phosphate buffered saline [PBS]), 0.04, 0.4, or 4.0 mg/kg anti-P2X4R receptor scFv95 (or scFv12 initial studies) was given intraperitoneally (i.p.) 3 weeks post-model induction. Naïve mice received vehicle injection. The potential for effectiveness of the P2X4R scFvs was tested in both male and female mice with FRICT-ION since nerve injury induced chronic neuropathic pain can be followed long enough to upregulate P2X4Rs on immune cells. As indicated by abundant literature, female mice were resistant to the treatment and thus the data beyond mechanical threshold testing are not shown for most tests. Animals were maintained on normal mouse breeder chow, which is lower in soy protein content and known to reduce inflammation and alter pain responses [70,71].

### 4.4. In Vivo Experimental Pain-Related Behavioral Testing

Lead P2X4R scFvs were selected with the best binding affinity and tested for efficacy to reduce pain-related measure in the FRICT-ION neuropathic pain model.

#### 4.4.1. Reflexive Mechanical Response Threshold Measurement Using von Frey Filaments

Hypersensitivity persists indefinitely in the FRICT-ION model, thus the model is suitable for assessing pain-like responses equivalent to the timeframe of chronic clinical pain (Montera & Westlund, 2020). Mechanical hypersensitivity was tested on the whiskerpad, the innervation territory of the infraorbital nerve, with von Frey filament stimulation at baseline and weekly thereafter as we have reported previously [32,72,73,74,75]. Reflexive responses to mechanical stimuli applied with graded von Frey filaments was tested weekly. A single trial consisted of 5 applications of several selected mid-range von Frey filaments applied once every 3 to 4 s. If no positive response is evoked, the next stronger filament is applied [76]. Responses to decreased gram force filaments indicated increased hypersensitivity (Figure 4).

#### 4.4.2. Anxiety- and Depression-like Behaviors

Cognitive-dependent behaviors are quantified once in week 8–10 after induction of the chronic model. Behaviors were video recorded for offline analysis.

Light/dark place preference test. Collected variables in this two chamber test were (1) time spent in each chamber, (2) number of transitions between chambers, (3) number of rearing events, and (4) entry latency into the light chamber [74,77,78,79].

Sucrose splash test. Depression-like behavior was tested with the sucrose splash test where decreased grooming behavior was defined as a measure of depression-like behavior [80,81]. Frequency, duration, and latency of grooming after spraying a 10% sucrose solution (~250 μL) on the base of tail were measured during the following 10 min.

### 4.5. In Vitro Electrophysiology

#### 4.5.1. Preparation of TG Cell Cultures for In Vitro Electrophysiology

TG were dissected from anesthetized mice and the resulting primary cell cultures utilized for electrophysiological and pre-block immunocytochemistry studies as previously [31]. After opening the skull and removing brain to expose the trigeminal nerve ganglia, the TG were removed from the connective tissue and dura mater at the base of the cranium. Ganglia were placed into a 35 mm dish containing 2 mL cold Hank’s Balanced Salt Solution (HBSS) without Ca^2+^/Mg^2+^, while keeping the dish on ice.

TG neurons were digested in an enzymatic solution containing sterile HBSS without Ca^2+^/Mg^2+^ (Gibco Cat# 14170112, Thermo Fisher, Waltham, MA, USA), papain (32.7 U/mgP, Cat# LS003126; Worthington, Lakewood, NJ, USA), L-Cysteine (Cat# C7352-25g, Sigma, St. Louis, Mo, USA), and saturated bicarbonate (Cat# S5761-500g, Sigma). This was followed by a secondary enzymatic digestion containing HBSS, dispase II (2 mg/mL, cat# D4693-1g, Sigma), and collagenase type 2 (2 mg/mL, cat# LS004176; Worthington, Lakewood, NJ, USA). After incubation for 20 min with gentle agitation under 37 °C for each enzymatic solution, TG were triturated for 45 s in complete Leibovitz’s L-15 medium (L-15 containing 5% fetal bovine serum, 1% antibacterial/antimycotic from 100 units/mL of penicillin, 100 µg/mL of streptomycin, 0.25 µg/mL of amphotericin B stock, and 2% of 1 M HEPES (cat# 11-415-064, Gibco). Then, the cells were layered onto complete L-15 medium and percoll gradient (density = 1.130 g/mL, 12.5% percoll layered over 28% percoll, cat# 89428-522, VWR, Radnor, PA, USA) and centrifuged for 10 min at 1300× *g*. Centrifugation was repeated in complete L-15 medium for 6 min at 1000× *g*. After the supernatant was carefully aspirated, the cell pellet was re-suspended in DMEM medium (10% fetal bovine serum and 1% antibacterial/antimycotic from 100 units/mL of penicillin, 100 µg/mL of streptomycin, 0.25 µg/mL of amphotericin B stock, cat# 11965, Thermo Fisher Scientific). Dissociated TG neurons were subsequently plated onto 12 mm poly-d-lysine-coated coverslips at a density of 1270 cells/mm^2^. Cultures were incubated 24 or 48 h in 37 °C and 5% CO_2_ until use.

#### 4.5.2. Whole Cell Patch-Clamp Electrophysiology

All recordings were performed 18–40 h following isolation as performed previously [31]. Neurons were identified by infrared differential interference contrast (IR-DIC) connected to an Olympus digital camera. Current clamp recordings were performed using a Molecular Devices Multiclamp 700B (Scientifica, Uckfield, UK). Signals are filtered at 5 KHz, acquired at 50 KHz using a Molecular Devices 1550B converter (Scientifica, UK), and recorded using Clampex 11 software (Molecular Devices, Scientifica, UK). Electrodes were pulled with a Zeitz puller (Werner Zeitz, Martinsreid, Germany) from borosilicate glass (GC150F, Sutter Instruments). Electrode resistance was 5–8 MΩ. Bridge balance was applied to all recordings. Intracellular solution contained (in mM) 125 K-gluconate, 6 KCl, 10 HEPES, 0.1 EGTA, 2 Mg-ATP, pH 7.3 with KOH, and osmolarity of 290–310 mOsm. Artificial cerebrospinal fluid (aCSF) contains (in mM) 113 NaCl, 3 KCl, 25 NaHCO_3_, 1 NaH_2_PO_4_, 2 CaCl_2_, 2 MgCl_2_, and 11 -glucose. Data acquisition was sampled at 20 kHz and filtered at 2.4 kHz. Recordings with a series resistance greater than 20 MΩ were discarded, and series resistance was compensated to 70%.

#### 4.5.3. P2X4R scFv Effects on Primary TG Culture Responses

The TG were derived from untreated mice with FRICT-ION in week 3. The effect of P2X4R scFv95 on neuronal excitability in large- (>25 μm) and small- to medium-diameter (<25 μm) neurons was determined. Resting membrane potential (RMP), rheobase, and spontaneous activity were recorded and quantified for group comparisons.

#### 4.5.4. Pre-Block of P2X4R scFv Immunostaining in TG Primary Cultures

Naïve and FRICT-ION male mice (*n* = 3) were euthanized, TG cells dissected and dissociated with enzymes (2 mg mL^−1^ dispase and 1 mg mL^−1^ collagenase IV (Worthington Biochemical, Lakewood, NJ, USA) at 37 °C). Primary cultures were established in 5% CO_2_ with D-MEM culture medium supplemented with 10% fetal bovine serum, 1% penicillin/streptomycin and 625 μmol L^−1^ glutamine, to stop the enzymatic reaction. Cells were plated and cultured for 48 h after isolation from naïve animals. Increasing concentrations of P2X4R scFv 95 (0, 0.5, 5, 10 µL) were added to each culture well (2.5 ug/mL in 1% FBS/BSA blocking buffer) to block P2X4R for 24 h (4 °C). Cells were fixed in 2% paraformaldehyde and blocked in 2% FBS/BSA blocking buffer (1 h) prior to overnight incubation in primary antibody (rabbit anti-P2X4R, 1:1000, APR-024, lot#APR024AN0225, Alomone Labs, Jerusalem, Israel). The cultures were incubated in secondary antibody (1:1000 donkey anti-rabbit AlexaFluor 488 (A21206 lot#1796375, ThermoFisher, Temecula, CA, USA) in 1% FBS/BSA blocking buffer for 1 h, rinsed with PBS, and mounted in Vectashield HardSet anti-fade with DAPI counterstain for imaging with a Leica TCS-SP8 fluorescent microscope. Quantitative analysis of immunostaining intensity of TG and pain pathway anatomical structures was performed in three rounds with over 120 cells per condition (30–50 cells per coverslip, 4–6 coverslips) using computer assisted fluorescence microscopy analysis in three rounds with a minimum of four regions/coverslip/animal to obtain mean staining intensities ± standard error of the mean (SEM) for group comparisons.

### 4.6. Western Blot

TG were collected during necropsy in week 10 and rapidly frozen for storage at −80 °C until protein content was assessed using standard Western blots methods. Briefly, the samples were denatured, proteins separated by molecular weight with gel electrophoresis, and electrotransferred on to a polyvinylidene fluoride membrane (PVDF). An anti-His-tag monoclonal antibody (C-terminal A01857, GenScript) that binds to the His-tag conjugated to the P2X4R scFv95 or anti-β-actin antibody (ab8226, Abcam) was applied to the PVDF membrane. Primary antibodies were visualized using appropriate secondary antibodies conjugated to fluorescent probes. Blots were imaged using a LICOR Odyssey Fc and the images analyzed using ImageJ.

### 4.7. Statistical Analysis

Study comparison groups included naïve, surgical sham, and nerve injured male BALBc mice with and without (w/wo) scFv. The power analysis predicted sufficient power is provided with group size *n* = 3, based on the pilot von Frey behavior data and mean in our previous assessments [8,42]. However, studies were repeated twice to produce behavioral data and fresh tissue samples (*n* = 6–8). All data passed normality tests (Shapiro–Wilk at minimum, alpha = 0.05). Two-way ANOVA (Dunnett’s multiple comparisons test) was performed on all von Frey behavioral data. One-way ANOVA (Dunnett’s multiple comparisons tests) was used to compare expression changes to controls in Western blot and histochemical analyses, as well as in comparisons of the anxiety- and depression-like behavior, conditioned place preference (CPP), and novel object recognition tests. In all cases, α = 0.05 was accepted for significant differences and data are expressed as the means +/− SEM for independent male and female group analyses. Curve fitting was performed in GraphPad Prism 8.1.2 using a non-linear regression model. Significance and post hoc analyses are shown in each study figure.

## 5. Conclusions

Ribosome display is a powerful cell-free technology and this technology is widely used to select single-chain antibody fragments against the target of choice due to reduced self-immunogenicity as well as easy and inexpensive large-scale production. This rapid method was used to quickly develop repertoires of high-affinity antibodies targeting P2X4R for our studies. In our hands, the scFvs developed with ribosome display have higher affinity, superior stability and solubility. Their small size has the potential for reduced self-immunogenicity. The innovation of this project also includes for the first ever demonstration of permanent reversal of chronic neuropathic pain-related behaviors by single-dose administration of P2X4R scFv. Use of scFv in TG cell cultures demonstrated a reduction in neuronal P2X4R expression on primary sensory neurons themselves. The pre-block study indicated at least part of this effect can occur directly on the trigeminal neurons themselves. Western blot demonstrated the his-tag localization in TG, medulla and amygdala indicating that the P2X4R scFv crossed the blood–brain barrier and entered the peripheral trigeminal nerve and brain. The project findings showcase this new avenue for development of non-opioid therapeutic interventions for chronic pain.

## 6. Patents

17/284208 PCT/US2019/059366 Non-Provisional: Therapeutic Antibody Fragments, Making, and Methods of Use. Pub. No.: US 2021/0340265 A1.

File 0310.000152WO01 Provisional: ScFv Antibody Block of P2X4R for the Treatment of Chronic Pain.

## Figures and Tables

**Figure 1 ijms-22-13612-f001:**
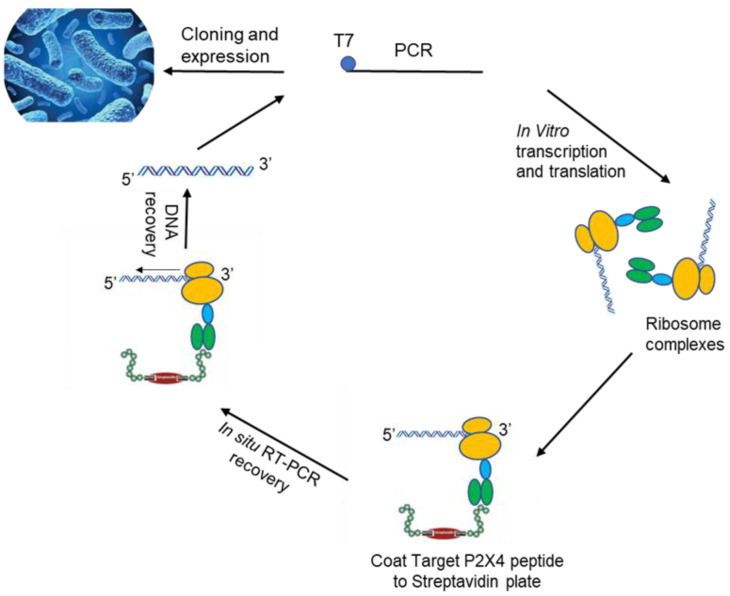
Overview of eukaryotic ribosome display selection, isolation and efficacy of anti-P2X4R scFv antibodies. Schematic of stalled ARM complex and position of primers used for RT-PCR recovery in the first, second, and third cycles of ribosome display. The resulting library was converted to ribosome display format for transcription to mRNA, translation, and selection.

**Figure 2 ijms-22-13612-f002:**
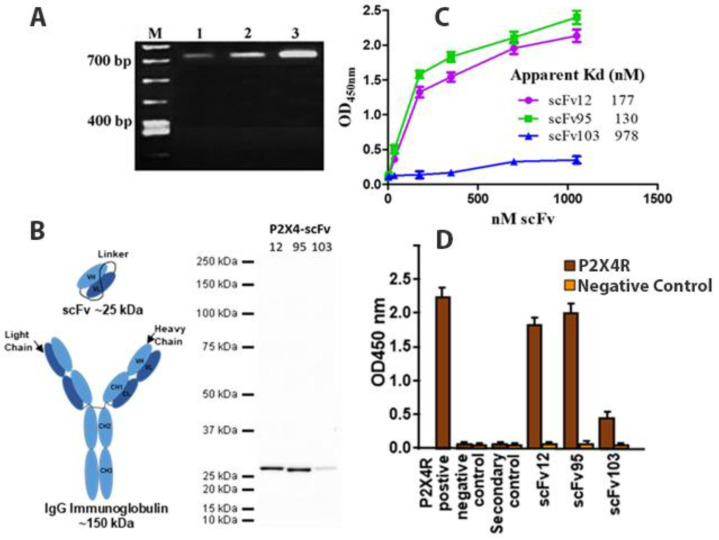
Isolation and binding efficacy of P2X4R scFv antibodies. (**A**). Analysis of RT-PCR recovery of V_H_/K cDNA from P2X4R peptide-immunized spleen in the 1st, 2nd, and 3rd cycles. (**B**). The diagram to the left illustrates the size difference between scFv and full IgG molecules. On the right are Western blots of three purified unique P2X4R scFvs generated by the cell-free ribosome display platform. (**C**). Plot of ELISA data showing nanomolar binding affinity for the three scFv antibodies with the best binding. (**D**). Binding specificity and cross-reactivity of three P2X4R scFvs to P2X4R, but not to neutral IgGs or scFvs as negative controls.

**Figure 3 ijms-22-13612-f003:**
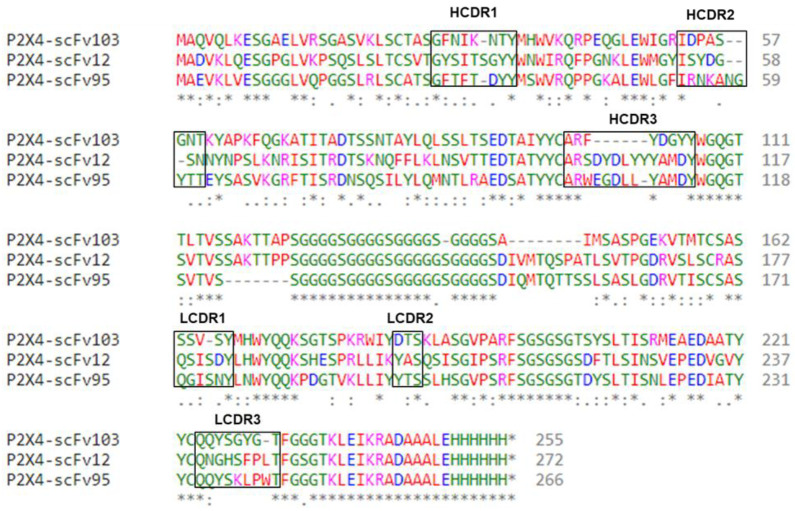
Amino acid sequences of V_H_-Linker-V_L_ of P2X4R peptide. Amino acid sequences of V_H_-Linker-V_L_ of P2X4R peptide-specific three mouse scFvs using Clustal Omega. Structural framework regions (FRs) and Complementarity-determining regions (CDRs) are determined by the IMGT information system. Diversity was found predominantly in the CDR regions. A normal 20 amino acid linker [(G4S)4] joins the V_H_ and V_L_ chains. Alignments were color coded according to residue property groups. AVFPMILW—red, DE—blue, RK—magenta, STYHCNGQ—green, and others—grey. * indicates sequence homology.

**Figure 4 ijms-22-13612-f004:**
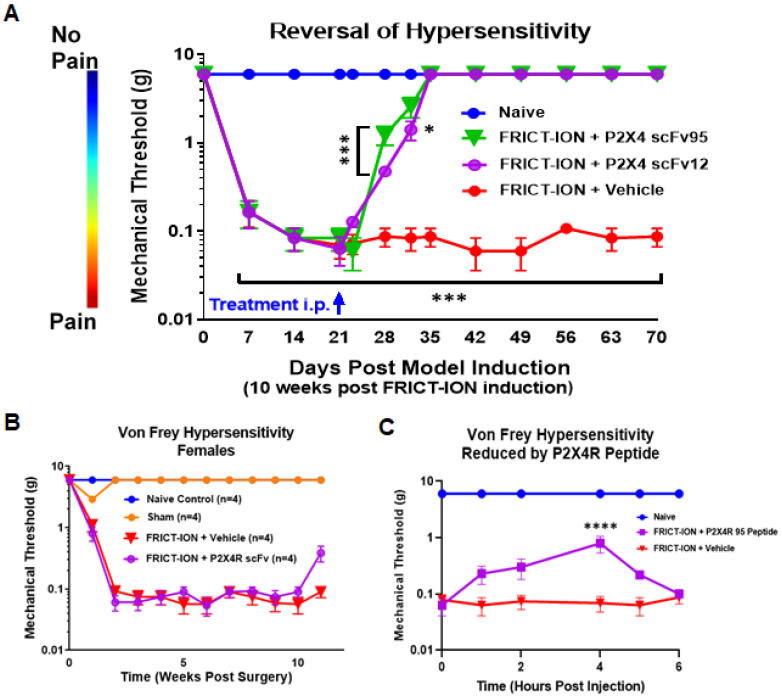
P2X4R scFv reverses mechanical hypersensitivity in male mice. (**A**). A single dose of two P2X4R scFvs (95 or 12) given 3 weeks after induction of FRICT-ION model permanently reverses mechanical hypersensitivity in male mice through to the end of the seven week study time course (*n* = 6, two-way ANOVA (Dunnett’s multiple comparisons test)). (**B**). The scFv had no effect on responses of female mice with FRICT-ION-induced mechanical hypersensitivity (*n* = 4). (**C**). Blocking peptide targeting the same P2X4R peptide fragment significantly reduced hypersensitivity, increasing the mechanical threshold 4 hours after treatment but the effect was short-lived. * *p* < 0.05, *** *p* < 0.001, and **** *p* < 0.0001.

**Figure 5 ijms-22-13612-f005:**
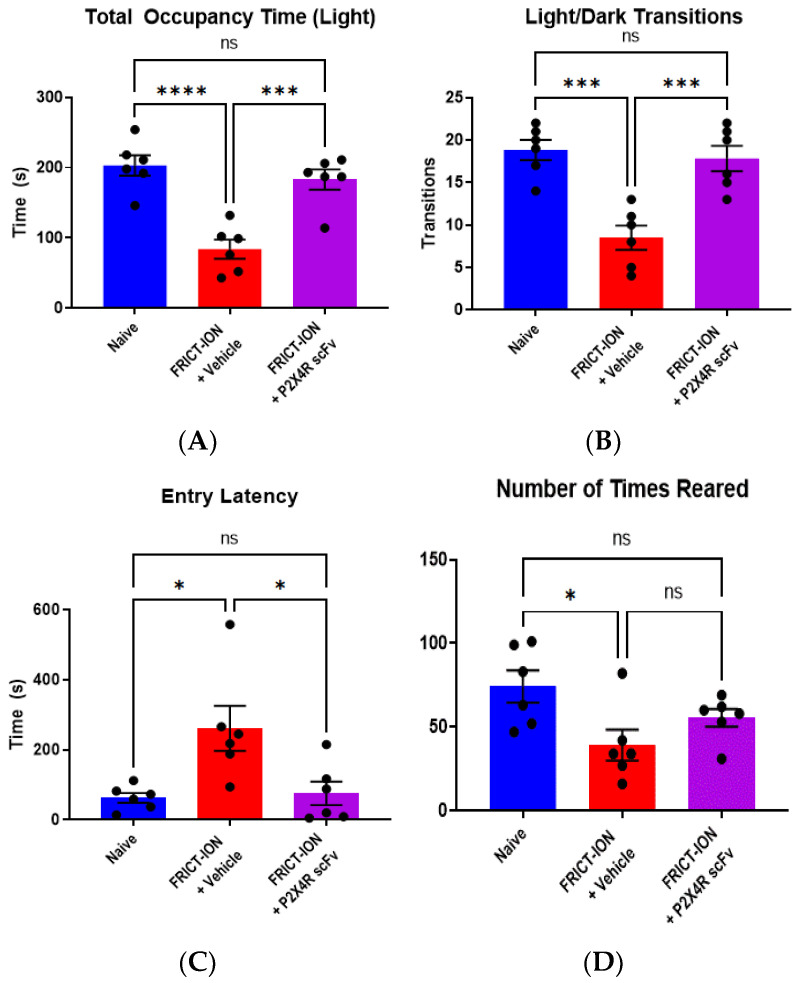
Anxiety is prevented by P2X4R scFv. The light/dark place preference test indicated all anxiety-like measures were significantly altered in FRICT-ION mice. P2X4R scFv95 prevented development of (**A**) time spent in the darkened box, (**B**) reduced number of transitions between the boxes, (**C**) decreased latency for lighted box entry, and (**D**) decreased number of rearing behaviors. *n* = 6, one-way ANOVA (Dunnett’s multiple comparisons test). * *p* < 0.05, *** *p* < 0.001, and **** *p* < 0.0001.

**Figure 6 ijms-22-13612-f006:**
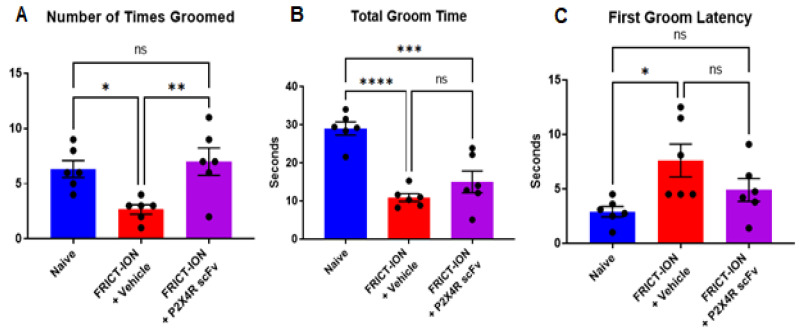
P2X4R scFv prevents development of depression. FRICT-ION model alters all grooming measures revealed with the Sucrose Splash test. (**A**). Number of grooming events is significantly increased by scFv95 compared to vehicle-treated FRICT-ION mice. (**B**). Treatment with P2X4R scFv95 does not alter time spent grooming in FRICT-ION mice, (**C**). but latency until the first grooming event is not significantly different compared to naïve controls. *n* = 6, ns = not significant, one-way ANOVA (Dunnett’s multiple comparisons test). * *p* < 0.05, ** *p* < 0.01, *** *p* < 0.001, and **** *p* < 0.0001.

**Figure 7 ijms-22-13612-f007:**
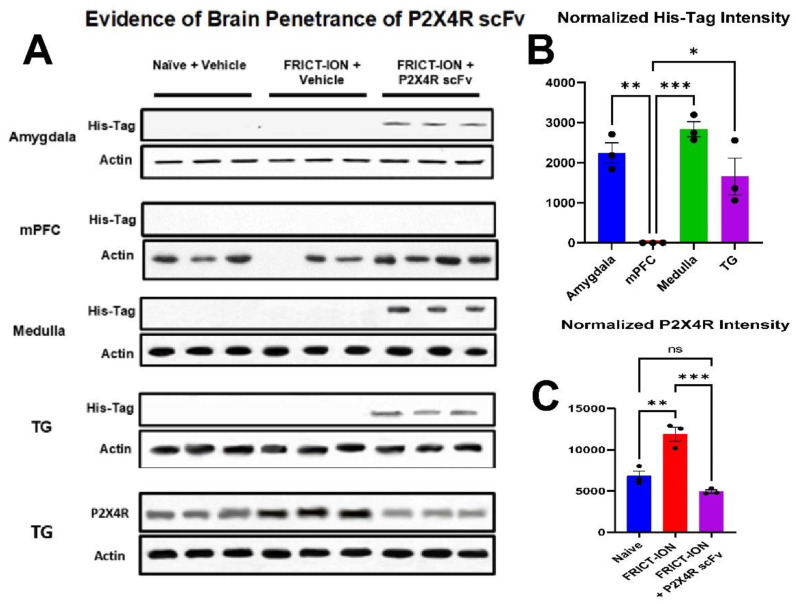
Evidence of P2X4R scFv 95 in brain and trigeminal ganglia, while P2X4R scFv 95 treatment prevents the P2X4R protein elevation in trigeminal ganglia of untreated FRICT-ION mice. (**A**). His-tag biomarker in the brain indicates brain penetrance of the P2X4R scFv 95 in amygdala, medulla, and trigeminal ganglia. (**B**). Bar graph of the scFv His-tag in the amygdala, brainstem medulla dorsal horn, and trigeminal ganglia (TG) in the Western blots. The His-tag remains at 7 weeks after the single i.p. dose was given in male mice with FRICT-ION. Almost no His-tag was detected in the medial prefrontal cortex (mPFC) in these comparisons with normalized intensity in arbitrary units. (**C**). The P2X4 protein increased in trigeminal ganglia of FRICT-ION mice with neuropathic pain (week 10) was not evident in mice treated with P2X4R scFv 95. *n* = 3, One-way ANOVA, * *p* < 0.05, ** *p* < 0.01, *** *p* < 0.001. *n* = 3, one-way ANOVA. * *p* < 0.05, ** *p* < 0.01, and *** *p* < 0.001.

**Figure 8 ijms-22-13612-f008:**
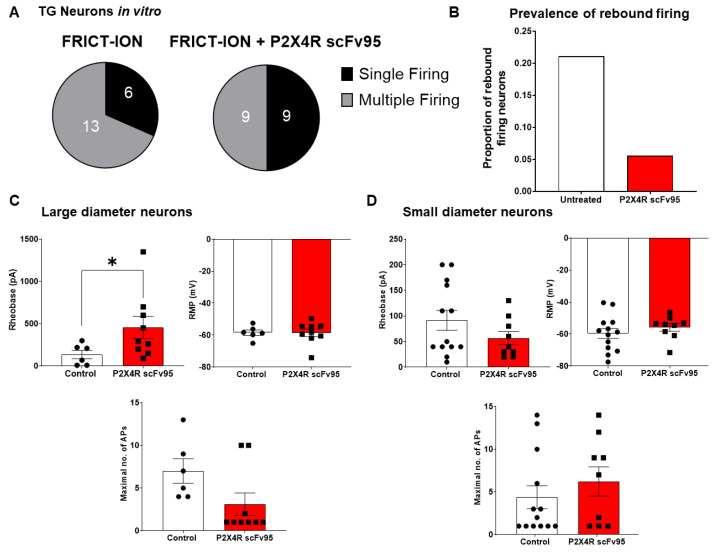
Effect of in vitro P2X4R scFv95 treatment on electrophysiological properties of TG neuron primary cultures from FRICT-ION mice. (**A**). Distribution of single vs. multiple action potential (AP) firing from untreated FRICT-ION and P2X4 scFv95 pre-treated FRICT-ION TG neurons. Prevalence of multiple firing was higher in untreated TG neurons from FRICT-ION mice than those with P2X4R scFv95 pre-treatment. (**B**). Prevalence of rebound firing in untreated FRICT-ION and P2X4R scFv95 pre-treated FRICT-ION TG neuronal cultures. Neurons with more rebound firing were observed more often in untreated cultures than in P2X4R-scFv-treated TG neurons from FRICT-ION mice. Effect of P2X4R-scFv treatment on electrophysiological properties of (**C**). large-diameter TG neurons (>25 microns) and (**D**). small- to medium-diameter TG-neurons (<25 microns). Untreated control *n* = 19 neurons, 8 mice. P2X4R scFv-treated *n* = 18 neurons, 5 mice. P2X4R scFv95 was used at a concentration of 4.5 μg/ml and treatment was performed for 1–2 h in the culture media. RMP = resting membrane potential. AP = action potential.

**Figure 9 ijms-22-13612-f009:**
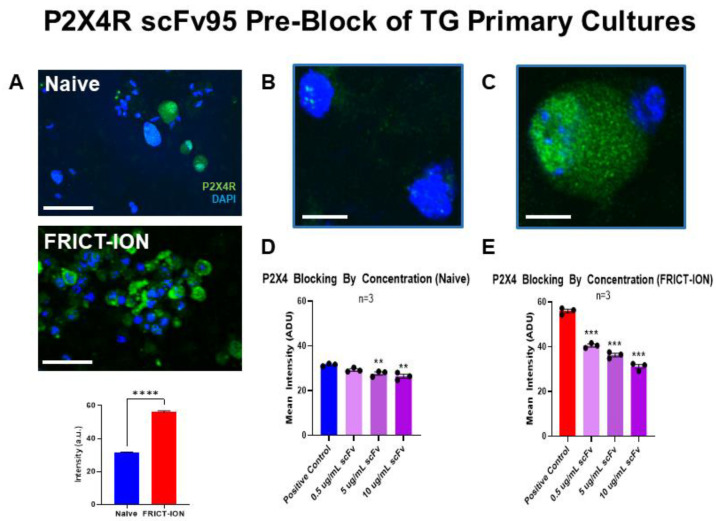
Pre-block of TG neurons with P2X4R scFv95. Low-power images of neurons immunostained for P2X4R (green) are shown in (**A**) for TG isolated from naïve and FRICT-ION mice. The bar graph of arbitrary staining intensity indicates the significant increase P2X4R in TG of mice with FRICT-ION compared to naïve. (**B**). At high power minimal P2X4R is found sequestered in vesicular structures in the TG of naïve mice. (**C**). Abundant staining for P2X4R is found in TG neurons isolated from FRICT-ION mice. (**D**). Pre-block with scFv 95 (0.5, 5,10 µg/mL, 24 h) significantly diminished P2X4R staining intensity in TG from naïve mice. (**E**). TG cells from mice with FRICT-ION pre-treated with P2X4R scFv95 had significantly diminished P2X4R staining intensity with all pre-block dilutions tested. All images are shown with DAPI blue counterstain. scFv. *n* = 4. ** *p*< 0.01, *** *p*< 0.001, and **** *p*< 0.0001. Bar = 50 µm in (**A**) and 6 µm in (**B**,**C**).

## Data Availability

Not applicable.

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
