# Peer review of "Single-Dose P2 X4R Single-Chain Fragment Variable Antibody Permanently Reverses Chronic Pain in Male Mice"

_ijms, 2021, doi:10.3390/ijms222413612_

Round 1

Reviewer 1 Report

This manuscript describes a series of studies to validate an antibody-based therapeutic that blocks P2X4R receptors and appears to profoundly alter and reverse the course of chronic pain in males. The scientific rationale for the study is logical and well explained. The animal model chosen for the behavioral studies showing elimination of pain is appropriate, as are the methods used to characterize the functionality of the antibody. The statistical methods are also generally applicable to the study design and able to provide convincing evidence for the authors' conclusions. Overall this is an innovative and promising study and may result in important breakthroughs in treating chronic pain. 

Specific critiques
1. Figure 2: This figure should be cleaned up. In B, the western blot, the caption indicates 8 antibodies are shown, when in fact only the best 3 are shown. Section D appears to have the legend partially covered and is not easily interpreted.
2. Figure 3 is referred to as Supplemental Fig. 1 in the text.
3. Figure 4, some data points seem to be missing error bars. 
5. The Figure 8 letters don’t match the text: B should be rebound firing, C large diameter, D small diameter.
6. Figure 9 FRICT-ION images don’t have a scale bar.

Author Response

This manuscript describes a series of studies to validate an antibody-based therapeutic that blocks P2X4R receptors and appears to profoundly alter and reverse the course of chronic pain in males. The scientific rationale for the study is logical and well explained. The animal model chosen for the behavioral studies showing elimination of pain is appropriate, as are the methods used to characterize the functionality of the antibody. The statistical methods are also generally applicable to the study design and able to provide convincing evidence for the authors' conclusions. Overall this is an innovative and promising study and may result in important breakthroughs in treating chronic pain.
Specific critiques 1. Figure 2: This figure should be cleaned up. In B, the western blot, the caption indicates 8 antibodies are shown, when in fact only the best 3 are shown. Section D appears to have the legend partially covered and is not easily interpreted.
We have fixed this issue.
2. Figure 3 is referred to as Supplemental Fig. 1 in the text.
We have fixed this issue.
3. Figure 4, some data points seem to be missing error bars.
These data points are not missing error bars. The error bars are extremely small and are not visible at the scale on the figure. We cannot adjust this.
5. The Figure 8 letters don’t match the text: B should be rebound firing, C large diameter, D small diameter.
We have fixed this issue.
6. Figure 9 FRICT-ION images don’t have a scale bar.
We have fixed this issue.

Reviewer 2 Report

The article is very interesting. The results are very clear and proves the effect of P2X4R scFv antibodies on chronic pain models

I only have one small suggestion. Please re edit Figure 8 and 9 with better resolution file. The text and labelling especially in Figure 8 is hard to read. Thank you 

Author Response

The article is very interesting. The results are very clear and proves the effect of P2X4R scFv antibodies on chronic pain models
I only have one small suggestion. Please re edit Figure 8 and 9 with better resolution file. The text and labelling especially in Figure 8 is hard to read. Thank you.
We have provided high resolution TIF files that can be downloaded separately along with the manuscript file. We hope that the journal will address this in the final publication.

Reviewer 3 Report

From the examination of the file that I have downloaded, unfortunately it seems to me that some figures mentioned in the text seem to be missing and also the current order is wrong. I cannot therefore proceed with the revision. I don't understand what happened and in any case I invite the authors to review the manuscript and submit it again.

Author Response

From the examination of the file that I have downloaded, unfortunately it seems to me that some figures mentioned in the text seem to be missing and also the current order is wrong. I cannot therefore proceed with the revision. I don't understand what happened and in any case I invite the authors to review the manuscript and submit it again.

The correct manuscript file was provided again.